# Nanofiber-Based Odor-Free Medical Mask Fabrication Using Polyvinyl Butyral and Eucalyptus Anti Odor Agent

**DOI:** 10.3390/polym14204447

**Published:** 2022-10-20

**Authors:** Jean-Sebastien Benas, Ching-Ya Huang, Zhen-Li Yan, Fang-Cheng Liang, Po-Yu Li, Chen-Hung Lee, Yang-Yen Yu, Chin-Wen Chen, Chi-Ching Kuo

**Affiliations:** 1Institute of Organic and Polymeric Materials, Research and Development Center of Smart Textile Technology, National Taipei University of Technology, Taipei 10608, Taiwan; 2Institute of Polymer Science and Engineering, Advanced Research Center for Green Materials Science and Technology, National Taiwan University, Taipei 10617, Taiwan; 3Division of Cardiology, Department of Internal Medicine, Chang Gung Memorial Hospital-Linkou, College of Medicine, Chang Gung University, Taoyuan 33305, Taiwan; 4Department of Materials Engineering, Ming Chi University of Technology, New Taipei City 24301, Taiwan

**Keywords:** medical mask, essential oil, electrospinning, polyvinyl butyral, safety

## Abstract

Following the 2020 COVID-19 worldwide outbreak, many countries adopted sanitary and safety measures to safeguard public health such as wearing medical face mask. While face masks became a necessity for people, disadvantages impede their long period wearing such as uncomfortable breathability and odor. The intermediate layer of the medical face mask is composed of porous non-woven fabric to block external particles while maintaining breathability. To overcome aforementioned limitation, this study uses electrospinning to design and fabricate odorless face masks via the use of aromatic oil. Eucalyptus essential oil is encapsulated through mixing and layer-by-layer by hydrophobic polyvinyl butyral and further used to fabricate the medical mask intermediate layer. We found that adding 0.2 g of eucalyptus into polyvinyl butyral fabric through mixing results in the deodorization rate of 80% after 2 h, with fabric thickness of 440.9 μm, and melt-blown non-woven fabric thickness of 981.7 μm. The Particle Filtration Efficiency of 98.3%, Bacterial Filtration Efficiency above 99.9%, and the differential pressure of 4.7 mm H_2_O/cm^2^ meet the CNS 14774 standard on medical face masks. Therefore, this study successfully proved that this type of masks’ middle layer not only effectively protects against coronavirus, but also provides better scents and makes it more comfortable for consumers.

## 1. Introduction

Following the 2020 COVID-19 outbreak worldwide, a wide majority of countries adopted sanitary and safety measures such as social distancing and face masks to safeguard public health. For instance, the World Health Organization (WHO) sketched a recommendation guideline on the 5 June 2020 to extend and widen the use of medical masks with multilayer textile fabrics to mitigate the propagation of COVID-19-associated aerosol and droplets [1,2]. Commonly, medical masks are layered with a porous polymeric non-woven fabric sandwiched by external electrospun non-woven fabric layer and inner melt-blown fabric layer [3,4,5,6,7]. Mask safety relies on the blockage of external particles, droplets, and virus within hydrophobic, charged, fibers or porous textile fabrics [8]. The presence of pores on the fabric surface enables external agent adsorption via the formation of intermediate Van der Waals interaction [9] or electret treatment [10] to effectively suppress infection hazards. However, user’s breath, atmosphere moisture, or high hygrometry environment induce the dissipation of the static electricity, resulting in significant protection efficiency loss [11,12,13,14]. Therefore, a novel mask fabrics design with hydrophobic polymer is required to decrease static electricity loss while providing breathability and high foreign agent filtration efficiency.

Electrospinning was recently used for a wide range of application designed by polymeric nanofibers fabric due to their unique physical and chemical properties [15,16,17,18]. Significantly, nanofibers fabrics have a very high specific surface area to volume ratio, form a highly porous network structure and developed interpore connection while maintaining high filtration and low air resistance [19,20]. Over the past few years, with the rapid development of nanofibers synthesis and characterization, research endeavors aimed to develop their potential for energy generation and storage, water treatment, sensing, healthcare, medical mask, and biomedical engineering applications [7,21,22,23]. In order to prepare versatile fibers, polymeric materials have widely been used due to their flexible mechanical properties [24]. For instance, polypropylene (PP) was employed in medical masks due to its porous structure [6]. Wang et al. employed polyvinyl alcohol (PVA) in order to extend the active mask layer static electricity capabilities [3]. They highlighted that the presence of hydroxyl group favor charge retention through formation of H-bond with neighboring water molecule, results in enhanced triboelectric effect. Zhu et al. added polyethylene glycol (PEG) to PVA in order to enhance mask rejuvenation properties via the exploitation of TiO_2_ particle photocatalytic properties [5]. Jimenez et al. used polyethylene terephthalate (PET) to fabricate surgical masks as a way to provide a second life to plastic waste, the resulting mask presented highly protective properties [25]. Polyvinyl butyral (PVB) was employed as a polymeric fabric for intermediate mask layer fabrication [26] due to its antibacterial capabilities [27]. Smart designs around polymeric fabric have attracted wide attention due to their flexibility and high active surface. Although the precise design principles of the multi-layered masks to limit the spread of COVID-19 have been well-documented, numerous combinations of materials and fabrics that yield a large variability in terms of comfort, filtration, and breathability exist. However, the aerosol capture does not prevent smell generation that reduces mask wearing comfort and needs to be addressed.

The principle of sensory deodorization relies on passive deodorization and can be divided into the following three types: aromatic species, masking species, and neutralizing species that include volatile vegetable oils [28,29]. All odors’ sources are volatile chemicals that have different structures, by extension different smell. Furthermore, smell sense is the only sense to directly transmit smell stimulation to the brain; smell is associated with memory, resulting in smell perception to be familiar or emotional [30]. In order to suppress unpleasant odors, a substance with high adsorption performances or high decomposability mixed into the fiber or used as a processing agent to remove the fabric odor is required [31]. The deodorization may, through physical adsorption for a highly porous structure, activate intermolecular Van Der Waals interaction and adsorb odor within pores [32,33]. Physical adsorption can effectively remove a wide range of odorous substance, although it cannot achieve an ideal removal effect due to exceeding the saturated state. Therefore, it needs to be exposed to various sources to remove the odorous substance from the pore before reuse [34]. Over the decades, numerous clinical experiments and studies have confirmed that the aroma of essential oils makes people simultaneously calm and relaxed [35,36]. For instance, Lehrner et al. [37], demonstrated that changing the ambient odor from citrus essential oil to lavender essential oil result in lowered anxiety levels with calmness and positive emotion. Most essential oils are medically used to recover olfactory loss [38]. In addition, eucalyptus oil possesses a high antimicrobial activity alongside various benefits for the human body [39,40,41]. Similarly to others essential oils not only is eucalyptus used as an anti-odor agent with benefits for human health but it is also employed to reduce and combat COVID-19-induced smell loss [42]. Domingues et al., demonstrated that various essential oils, including eucalyptus oil, can be very effective against Escherichia Virus MS2, a surrogate of SARS-CoV-2, resulting in increased mask multifunctionality [43]. Therefore, novel medical mask fabrics are required to exhibit high breathability, permeability, high particle filtration efficiency, as well as efficient odor removal to improve medical mask usage.

Herein, we use PVB due to its non-toxicity, porosity, hydrophobicity, and environmentally friendly properties, combined with eucalyptus to develop two encapsulation approaches, namely mixing and layer-by-layer. PVB high porosity enables highly efficient molecule absorption and desorption as well as antibacterial performances. Eucalyptus oil was chosen due to its beneficial odor, antimicrobial properties, and relaxing effect to improve mask user comfort. The resulting fabric is aimed to be incorporated into medical masks as an intermediate layer. The Fourier Transform Infrared (FTIR) characterization of the mixing fabric confirms that the eucalyptus microcapsules are well-embedded into the PVB nanofibers. Moreover, we found that adding 0.2 g of eucalyptus microcapsules via mixing and 0.5 g via layer-by-layer to nanofiber membranes makes the deodorization rate achieve 80%. After fabricating medical face masks using mixing and layer-by-layer fabric, the Particle Filtration Efficiency (PFE) reaches 98.3% and 99.8% respectively. The Bacterial Filtration Efficiency (BFE) is above 99.9% for both fabric while the mixing differential pressure is 4.7 mm H_2_O/cm^2^ meeting the CNS 14,774 standard on medical face masks. Therefore, this study successfully proves that this type of masks’ middle layer not only effectively protects against external particles such as coronavirus droplets, but also provides better long-term scents diffusion to increase user’s safety and comfort.

## 2. Experimental Part

### 2.1. Chemicals

Highly viscous PVB was purchased from Changchun Chemical, B-18HX. Eucalyptus microcapsules with average size ranging from 3 to 12 um, coated with melamine shell for good resistance and dispersibility, were purchased from Chongyu Technology Co., Ltd., Shenzhen, China; 95% ethanol was purchased from Jingming Chemical Co., Ltd., Tianjin, China. All products were used as received.

### 2.2. Experimental Equipment and Instrumentation

Stainless steel needle electrospinning equipment specifications are as follows: No. 18: inner diameter: 0.96 mm, outer diameter: 1.26 mm; length: 25.4 mm. No. 22: inner diameter: 0.42 mm, outer diameter: 0.72 mm; length: 2.54 mm. All needles were supplied by Hong Zhun Enterprise Co., Ltd. (Taiwan). The needle was placed at a vertical position on the electrospinning set-up with the following: high voltage power supply (manufacturer SIMCO Industrial (Taiwan)), Static Control Regulator (LOKO power (Taiwan)), and the injection pump (kd Scientific (Holliston, MA, USA)).

### 2.3. Electrospun Fabric Preparation

The PVB was dissolved in ethanol at 8 wt% under stirring 200 rpm at 70 °C until complete dissolution. For the preparation of PVB nanofiber membrane mixed with eucalyptus microcapsule via mixing approach, eucalyptus microcapsules (from 0.1 to 0.2 g by 0.025 g increment) were added into 3 mL of PVB solution followed by 1 h stirring until complete microcapsule dispersion, followed by electrospinning. The electrospinning was conducted with the following parameters, the voltage was set at 13.8 kV with the collector plate placed at 15 cm from the spinneret with a flow rate of 1.0 mL per hour, the needle was 0.96 mm wide, at last the temperature was set at 25 °C (±5) and humidity at 40% (±5). For the layer-by-layer approach, 5 mL of the water-dissolved eucalyptus microcapsules (from 0.1 to 0.5 g by 0.1 g increment) were sandwiched between electrospinning of 0.75 mL of PVB solution fiber membranes. The process was repeated 4 times with the following electrospinning parameters, the voltage was set at 13.8 kV with the collector plate placed at 15 cm from the spinneret with a flow rate of 1.0 mL per hour, the needle was 0.72 mm wide, at last the temperature was set at 25 °C (±5) and humidity at 40% (±5).

### 2.4. Characterization

Resulting nanofibers fabrics were characterized as follow by Scanning electron microscope (SEM) (JSM-6510, JEOL) and FTIR spectrometer (Nicolet iS50 FT-IR) was provided by Thermo Scientific. Fibers Specific Surface Area and Porosimetry was conducted by a Porosimetry Analyzer (ASAP 2020 V3.04, Micromeritics). The fabric air permeability test was conducted using a FX 3300 Air Permeability TESTER III. Gas Sampler (GV-100S, Gastec) was used to be put sample into a detector tube (Model: 105SE, Manufacturer: Sensidyne. Automatic Canister Tester, Model: 8130A, Supplier: TSI). Further, the exchange pressure used was manufactured by the Textile Industry Research Institute and the Bacterial Filtration Test Kit was provided by the Textile Industry Research Institute.

### 2.5. Deodorization Test

The deodorization test was conducted according to the ISO 17299-22014 procedure. A detection tube with an accuracy error of less than 5% was used while the ammonia (Itabashi Gas Co., Ltd., Tokyo, Japan) with a detection range between 0.2 uL/L and 200 uL/L was stored within gas sampling bag (10 L content). A peristaltic pump was used to add ammonia until a concentration of 100 ppm, followed by gas collector extraction up to 100 mL. The sample was stored at 20 °C (±2) with a relative humidity of 65% (±4) for at least 24 h. Simultaneously, we prepared six 10 L gas sampling bags, with three for the test group and three for the control group. The three test samples preparation with an area of 100 (±5) cm^2^ were stored into a gas collection bag further rolled up until a volume of 5 L was reached. Then, the 6 gas collection bags were vacuumed with a peristaltic pump and further placed into 3 L of ammonia gas and stored in an environment at 20 °C (±2) with a relative humidity of 65% (±4) for at least 2 or 24 h. Upon waiting for 2 or 24 h, the gas collector was used to install the detection tube to extract 100 mL of gas from collection bag. Following gas extraction through the detection tube, a discoloration reaction value was produced and indicated the residual odor (ammonia) concentration left in the gas collection bag after contact time between test sample and ammonia. The resulting ammonia concentration was calculated in ppm and provided in Table 1.

### 2.6. Air Permeability Test

The test was conducted according to the CNS 5610 L3080-1980 non-woven inspection method and CNS 5612 L3081-2012 fabric air permeability test method. The sample tested area is of 20 cm^2^, the pressure difference was tested at 100 Pa on mixing and layer-by-layer fabric.

### 2.7. Air Exchange Pressure

The test was conducted according to the CNS 14,777 T4039-2003 test method for air exchange pressure of medical face masks. To follow the CNS 14,777 test method, the test was carried about in situ with no modification of the temperature and humidity of the chamber. The sample was placed on top of a 25 mm diameter hole to allow air to pass through the sample. After turning on the vacuum pump, the airflow was adjusted to 8 L·min^−1^ to monitor the differential pressure value.

### 2.8. Particle Filtration Efficiency

The test was conducted according to the test method CNS 14,755 Z2125-2011 for disposable dust mask, the main purpose was to measure the mask submicron protection efficiency. First, the sample was stored for 24 to 36 h within a box at constant humidity (85 ± 5%) and temperature (38 °C ± 2), followed by a 10 h machine test. A 2% NaCl solution was poured into an aerosol generator to reduce aerosol size to 0.3 µm. The air was flown through the mask at a flow rate of 85 L·min^−1^ (±4), followed by particle flow efficiency determination.

### 2.9. Bacterial Filtration Efficiency

The test was conducted according to the test method CNS 14,775 T4037-2003 for BFE of medial face mask, staphylococcus aureus bioaerosol was used to evaluate the BFE of medical mask [44]. The sample was cut into a 10 cm^2^ area and placed in an ordered manner into an environment at 21 °C ± 5 and 85% (±5) of humidity for at least 4 h. The test bacterial solution was prepared with the cultivation of staphylococcus aureus (ATCC 6538) into a test bacterial solution of 5 × 105 CFU·mL^−1^ with peptone aqueous solution. The injection pump was used to transport the bacterial liquid (particle size of 3.0 ± 0.3 µm) to the nebulizer with a bacteria delivery amount maintained at 2200 (±500) for each test. The aerosol collected from the sample positive control group was used to determine the delivered bacteria amount. Upon completion, the positive control group test was repeated with one control sample formed of every 13 groups followed by the addition of digested soybean agar (TSA) plate medium corresponding to number 1 to 6 sequence according to the Anderson sampler. Since the filtration efficiency of the aerosol was evaluated, the inner layer of the sample was facing toward the bacterial aerosol (upward) as required. The bacterial liquid was transferred to the nebulizer for 1 min, the high-pressure and vacuum pump were turned o for 2 min and the TSA was removed, then the cycle was repeated. Upon test completion, the Petri dish was incubated at 35 °C (±2) for 48 h (±4) followed by colony counting and converted into the actual bacterial count and further Bacterial Filtration Efficiency.

## 3. Result and Discussion

### 3.1. PVB—Eucalyptus Microcapsule Electrospinning

PVB is widely used in ink printing adhesives on food packaging in Europe and the United States as well as mask fabrication, ascribed to its remarkable flexibility and adhesion properties. Moreover, PVB has non-toxic and hydrophobic properties preventing the water vapor and the generated saliva from triggering the membrane mask degradation. The flexibility of PVB makes it a suitable candidate for electrospinning due to its high conductivity and surface tension. Briefly, PVB is dissolved in ethanol due to its environmental friendliness and while it can produce uniform fibers without the presence of droplets [45]. According to the preparation method mentioned in the experimental method, the PVB nanofiber membrane mixed with eucalyptus microcapsule was prepared via two methods: mixing [46] and layer-by-layer [47]. Eucalyptus microcapsules are organized in a core-shell structure with the eucalyptus essential oil being the core whereas the shell is composed of melamine. Several ratios of eucalyptus microcapsules (provided in the experimental section) were mixed with PVB. The mixing approaches ensure a strong encapsulation of material within fiber fabric and protection against degradation whereas the layer-by-layer approach warrant a uniform dispersion of eucalyptus microcapsules between PVB fiber layer. First, the eucalyptus microcapsules were dispersed alongside PVB at weights ranging from 0 to 0.2 g in the mixing layer. Second, the eucalyptus microcapsules were sprayed (after dissolution in water) alongside PVB at weights ranging from 0 to 0.5 g in the layer-by-layer approach. As shown in Figure 1, both mixing and layer-by-layer approaches are illustrated. Following the addition of 0.2 g of eucalyptus microcapsule during both fabrication approaches, the surface morphology was investigated. According to the Figure 2a–c SEM pictures associated to the layer-by-layer approach, the eucalyptus microcapsule dispersed within deionized water were sprayed directly on top of the PVB fibers without obvious crack with numerous inter-fibers contact points. The multilayer stacking suppresses the removal of microcapsule from the entangled network due to external environmental factor such as wind, and accidental touch [48]; thus, preserving the deodorization effect from weakening. Comparatively, Figure 2d–f SEM pictures show that eucalyptus microcapsules are directly embedded within PVB fibers, ascribed to the mixing fabrication approach relying on PVB–eucalyptus solution mixing. The advantage of this approach is a certain presence of microcapsule; however, the inequal capsule repartition of the fibers may cause an uneven film thickness and consequently degrade filtration performances.

### 3.2. PVB–Eucalyptus Microcapsule Nanofibers Membrane Analysis

We used FTIR to demonstrate the proper eucalyptus microcapsule encapsulation into PVB via the mixing approach. We observed PVB nanofibers membrane, pristine eucalyptus microcapsule, and mixing approach PVB–eucalyptus signal peak and we confirmed that the eucalyptus microcapsules were successfully embedded within the PVB fibers. As shown by Figure 3, the FTIR analysis of the PVB membrane showed an aliphatic C-H signal at 2906.5 cm^−1^ and a C-O-C signal peak at 1122.2 cm^−1^, the 1004.1 cm^−1^ peak is ascribed to the C-O stretching. For the eucalyptus microcapsule, in which the shell is composed of melamine, FTIR analysis is provided by 1335.4 cm^−1^ and 3340.4 cm^−1^ peaks ascribed to melamine amine group whereas 814.0 and 1559.2 cm^−1^ signal peaks are associated with the 1, 3, 5-triazine ring. Finally, the 2947.3 cm^−1^ signal peak is associated to the 1,8-cineole, the main component of the eucalyptus essential oil [49,50,51]. Further, mixing approach PVB–eucalyptus materials were analyzed by FTIR, the 1004.1 cm^−1^ C-O and 1139.1 cm^−1^ C-O-C stretching characteristic of PVB are visible. Upon encapsulation, all peaks associated with the melamine were present but weakened, indicating a strong encapsulation of the eucalyptus microcapsule by PVB. The triazine peaks at 1500 cm^−1^ in the FTIR plots almost disappeared which may be attributed to interaction with PVB.

### 3.3. Deodorization Performance Comparison

The deodorizing effect of microcapsule was investigated by mixing PVB–eucalyptus microcapsule fibers at different proportions through the mixing and layer-by-layer approach. According to Table 1, gas collection bags containing 100 ppm of ammonia were filled with PVB mixed with 0.1 g up to 0.2 g by 0.025 g increments of eucalyptus. After two-hour storage, the ammonia ppm concentration was reduced to 70, 60, 55, 42, and 20 ppm respectively. This shows that the nanofiber membrane filled with eucalyptus microcapsule can effectively eliminate ammonia via adsorption. Layer-by-layer samples (with 0.1, 0.2, 0.3, 0.4, and 0.5 g of eucalyptus microcapsules) were added into gas collection bags and upon two-hour storage, the ammonia ppm concentration fell to 80, 50, 40, 40, and 20 ppm, respectively. Furthermore, it was observed that upon addition of 0.1 g of eucalyptus through the mixing method, the deodorization rate was up to 30% according to Figure 4a and only 20% via the layer-by-layer approach. The addition of up to 0.2 g leads to an increase to 80% of deodorization rate via mixing whereas it remains at 50% when using the layer-by-layer approach. Significantly, via the mixing approach, we found that further eucalyptus addition does not affect the final deodorization rate remaining at 80%, thereby indicating that the mixing method adsorption reached an equilibrium point. Therefore, the mixing method allows to save microcapsule cost while achieving the best deodorizing effect in a shorter time. However, it is known that the substances encapsulated within the microcapsule are released following an external stimulus such as heat, friction, diffusion, pH change, or electric and magnetic field stimulation, favoring the destruction of the shell and release of the core agent at the vicinity of the fiber network [52]^.^ Hence, the long-term deodorization effect is weakened for the mixing approach, ascribed from the microcapsule cracked shell originating from the generated electric field during electrospinning [53]. It results in increased eucalyptus essential oil release kinetics and faster neutralization reaction with ammonia resulting in quick depletion of anti-odor agent. We found that after 24 h storage, the layer-by-layer deodorization rate is 92% and only 88% for mixing, originating from the controlled release of anti-odor agent of intact layer-by-layer eucalyptus microcapsules (Figure 4b). We found that optimal eucalyptus microcapsule loading is reached at 0.2 g for the mixing approach and 0.5 g for the layer-by-layer approach, both being at 80% deodorization rate. The optimized ratio is therefore used as the middle layer of the mask, and the melt-blown non-woven fabric middle layer of the medical mask is suitable to be replaced to make a PVB-based medical mask and conduct subsequent tests.

### 3.4. Adsorption and Desorption Analysis

We aim to replace the original medical mask melt-blown fabric with PVB–eucalyptus fabric; therefore, a wide range of tests are required to ensure proper breathability and protection alongside wearer comfort. The adsorption behavior of gas onto solid may be carried out via two ways, either physical or chemical adsorption. Considering physical adsorption, weak Van der Waals forces between gas molecules and solid surface enables adsorption of molecules within the polymer porous structure via activated carbon adsorption [54]. Therefore, desorption may occur by using a small amount of energy making the process reversible and non-dissociative while being capable of accommodating multiple layers of molecules. Herein, we employ nitrogen to determine the adsorption capabilities of our PVB fabric. A specific surface area and porosity analyzer was used to monitor pressure change before and after adsorption of the nitrogen by the interlayer melt-blown and PVB nanofibers membrane of the medical mask. The estimated adsorbed gas was extracted from the Brunauer–Emmett–Teller (BET) and Langmuir equation to calculate the specific surface area of our fabric [55]. Figure 5a,b describe the evolution of the nitrogen gas adsorption and desorption curve onto the PVB nanofibers and melt-blown fabric. According to the BET Formula (1), the V_m_ and C (σ) can be extracted upon knowing the cross-sectional area of the adsorbed molecule.
(1)$$\frac{P}{V_{a}\left(P_{0}-P\right)}=\frac{1}{V_{m}C}+\frac{C-1}{V_{m}C}\left(\frac{P}{P_{0}}\right)$$
where *V_a_* is the total volume of adsorbed gas at the pressure equilibrium, and *P* the adsorbed gas at equilibrium at the adsorption temperature. *P*_0_ is the saturation pressure of adsorbed molecule while *V_m_* is the amount of adsorption required to completely cover the entire surface with a monolayer of adsorbed molecules. *C* is a constant related to the material. By using the BET method, the total surface area and specific surface area of solid material can be calculated as shown by Equations (2) and (3) below [32]:(2)$$S_{BET,total}=\frac{\left(V_{m}N_{A}S\right)}{V}$$
(3)$$S_{BET}=\frac{S_{BET,total}}{a}$$
where *S_BET, total_* is the total of the surface area of solid material, *V_m_* is the volume of saturated adsorption capacity of monolayer and Na is Avogadro constant. S is the adsorption cross-sectional area of adsorbed species. *S_BET_* corresponds to the specific surface area of solid material while A is the mass of adsorbent material. Figure 5c,d display how the BET varies in accordance with the gas relative pressure and to determine the fabric specific surface area of the PVB and melt-blown fabric, respectively. The mass of the samples was fixed at 0.1 g. After the calculation, the specific surface area of the melt-blown non-woven fabric was determined to be 6.8907 m^2^·g^−1^ and the specific surface area of the PVB nanofiber membrane was 14.0375 m^2^·g^−1^. According to Equation (2), the specific surface area is proportional to the gas adsorption amount; therefore, the PVB nanofiber membrane gas adsorption capability is greater than the melt-blown non-woven fabric.

### 3.5. Fabric Thickness Comparison

Scanning Electron Microscopy (SEM) was used to determine the thickness of the melt-blown non-woven fabric intermediate layer of the medical mask and PVB nanofibers membrane via mixing (0.2 g eucalyptus) and layer-by-layer approach (0.5 g eucalyptus) (Figure 6a–c). The average thickness is measured as follows: the melt-blown fabric thickness is 981.7 µm, mixing and layer-by-layer fabric thickness is 440.9 and 676.9 µm, respectively. The thickness discrepancy is ascribed to the experimental fabrication approach; however, both approaches show that the resulting thickness is thinner than the melt-blown counterpart. Further, fibers diameter of the melt-blown non-woven fabric, mixing and layer-by-layer approach are shown in the SEM photograph (Figure 6d–f). Following measurement, the average fiber diameter of the mixing and layer-by-layer fiber diameter is 0.83 and 0.74 µm, respectively. Figure 6g shows that the melt-blown fiber diameter was unevenly distributed from 2 to 20 µm with an inhomogeneous thickness. By contrast, mixing and layer-by-layer fiber size distribution is provided in Figure 6h,i, respectively. We found that the mixing highly heterogeneous fiber size distribution is directly associated with the presence of encapsulated eucalyptus microcapsule. However, the average size range is similar to the layer-by-layer approach; therefore, considering a large area sample, the global thickness is homogeneous with imperceptible change to the air permeability of our fabric overall.

### 3.6. Air Permeability and Air Exchange Pressure

Air permeability refers to the rate at which air travels vertically through a fabric under a specific area, pressure differential, and time. The air permeability test was conducted according to CNS 5610 L3080-198 non-woven inspection method and CNS 5612 L3081-2012 fabric air permeability test method. The melt-blown non-woven fabric of the mask intermediate layer, the mixing and layer-by-layer fabric were prepared in a 20 cm^2^ surface area and put under pressure differential of 100 Pa. Following testing the air permeability, the layer-by-layer fabric air permeability was 13.7 mm·s^−1^, the mixing fabric air permeability was 28.6 mm·s^−1^, and the melt-blown air permeability fabric was 132.3 mm·s^−1^. Illustrated by Figure 7a, the melt-blown fabric air permeability is much higher, whereas layer-by-layer permeability is significantly low due to the spraying of eucalyptus microcapsule on a multilayer scale. We found that the fiber diameter has a significant role on the fabric air permeability capabilities, resulting in low fiber diameter reducing the air permeability. The air exchange pressure is the pressure drop toward the mask layers, which is an important parameter to evaluate the mask breathability and comfort. Test were carried according to the CNS 14,777 T4039-2003 procedure for mask air exchange procedure. The sample was placed in front of a 25 mm hole to allow air to go through the mask intermediate layer (composed of the melt-blown fabric, layer-by-layer or mixing fabric) and the test results are shown in Figure 7b. We found that the melt-blown fabric pressure is 4.8 mm H_2_O·cm^−2^, the pressure of mixing fabric is 4.7 mm H_2_O·cm^−2^ whereas layer-by-layer fabric pressure is 17.0 mm H_2_O·cm^−2^ which is far above the 5.0 mm H_2_O·cm^−2^ CNS 14,774 (T5017) specification for medical masks. By contrast, the mixing and melt-blown fabric approach are below the specification threshold. We conclude that the layer-by-layer approach enables high deodorization effect at the expense of the user comfort.

### 3.7. Bacterial Filtration Efficiency and Particle Filtration Efficiency

Commonly, masks are divided into general medical mask, surgical mask, and surgical D2 dust masks and classified in relation of their capabilities to block particle flow-through. In general, two major criteria are used to assess the general mask efficiency which is the differential pressure and BFE. Furthermore, the determination of the PFE determine the safety of a medical mask as the BFE is for 3.0 µm bacteria whereas the PFE is for 0.3 µm aerosol. According to CNS 14,775 T4037-2003 test method for BFE of medical face mask, we used the staphylococcus aureus bioaerosol following the performances specifications of CNS 14,774 (T5017) regulations. According to the CNS 14,774 (T5017) regulation, BFE should be greater than 95%. Simultaneously, the PFE is tested according to the CNS 14,755 Z2125-2011 test method for disposable dust masks and according to the CNS 14,774 (T5017) “medical face mask” to achieve surgical mask protection level, the PFE needs to be equal or higher than 80%. In order to achieve surgical D2 dust mask protection level, the protection is required to be equal or higher than 95%. The test results are shown in Figure 7c (BFE) and Figure 7d (PFE) and we observe that the BFE for all fabrics was above 99.9%. Simultaneously, layer-by-layer and mixing fabric, despite being thicker, exhibits low pressure difference (lesser than 5 mmH_2_O·cm^−2^) and above 95% BFE. Therefore, both fabrics are permitted to be integrated as a medical mask intermediate layer. PFE performances highlight that the melt-blown middle layer protection level is 91.9% for PFE and fails to meet the PFE criteria, whereas mixing and layer-by-layer intermediate layer protection reached 98.3 and 99.8% protection level, respectively, passing the surgical D2 protection level according to the 14,774 (T5017) medical face mask regulations and which is ascribed to the entangled fiber-capsule network. Our mask performances are summarized in Table 2 to emphasize their promising capabilities in comparison with current medical mask standards. To conclude, developed mixing and layer-by-layer PVB–eucalyptus fabric exhibits great potential as an intermediate layer for the development of future medical masks to upgrade public health and medical institution safety levels while preserving wearer well-being and safety.

## 4. Conclusions

Following the 2020 COVID-19 worldwide outbreak, medical face masks quickly became mandatory to wear; however, long-term mask wearing induces high discomfort. Herein, we not only combined a hydrophobic polymer PVB with an anti-odor agent, eucalyptus oil, but also exploited the advantageous electrospinning approach to fabricate highly efficient nanofiber fabric. Eucalyptus essential oil is encapsulated through mixing and layer-by-layer by hydrophobic and ecofriendly PVB and further used to fabricate the medical mask intermediate layer. We found that adding 0.2 g of eucalyptus into PVB fabric through mixing and 0.5 g through layer-by-layer approach results in deodorization rate of 80% after 2 h. The intermediate layer of the medical face mask is composed of porous non-woven fabric to block extern particle while maintaining permeability and breathability whereas eucalyptus diffuse to block poor odor to suppress mask wearing discomfort. The mixing and layer-by-layer PFE is 98.3% and 99.8%, respectively, and both fabrics’ BFE are above 99.9%. In addition, the mixing approach differential pressure of 4.7 mm H_2_O/cm^2^ meets the CNS 14,774 standard on medical face masks. Therefore, this study successfully proves that PVB–eucalyptus microcapsule via layer-by-layer or mixing to fabricate a face mask middle layer not only effectively protects against coronavirus, but also provides better scents, and makes it more comfortable for consumers.

## Figures and Tables

**Figure 1 polymers-14-04447-f001:**
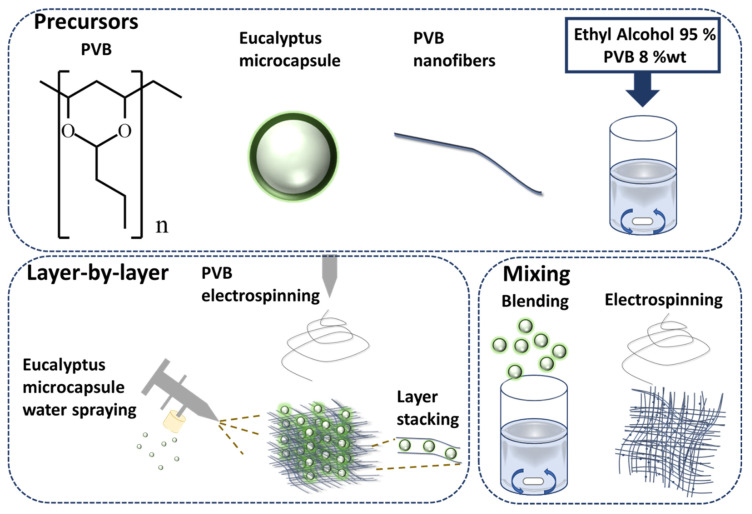
Mixing and layer-by-layer nanofibers fabric are fabricated as shown above. First, PVB is dissolved into ethyl alcohol. Second, via layer-by-layer approach, water-dissolved eucalyptus microcapsules are sprayed between PVB fibers stacking. Via mixing, eucalyptus microcapsules are mixed with PVB prior to electrospinning.

**Figure 2 polymers-14-04447-f002:**
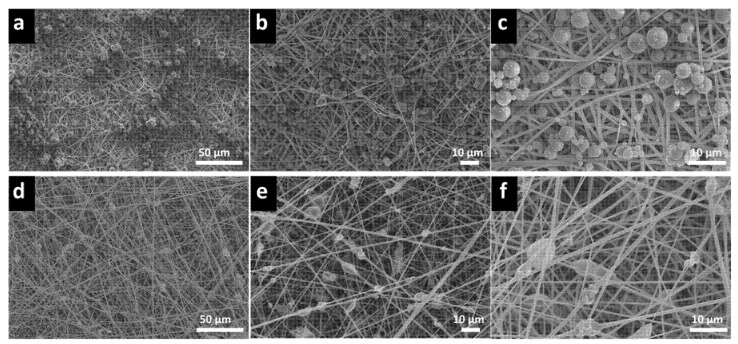
The SEM picture describe the incorporation of eucalyptus microcapsule within PVB nanofibers fabric. As shown in (**a**) (×500), (**b**) (×1000) and (**c**) (×2000), eucalyptus microcapsules are intercalated through layer-by-layer. As shown in (**d**) (×500), (**e**) (×1000) and (**f**) (×2000), eucalyptus microcapsules are embedded into PVB nanofibers via mixing.

**Figure 3 polymers-14-04447-f003:**
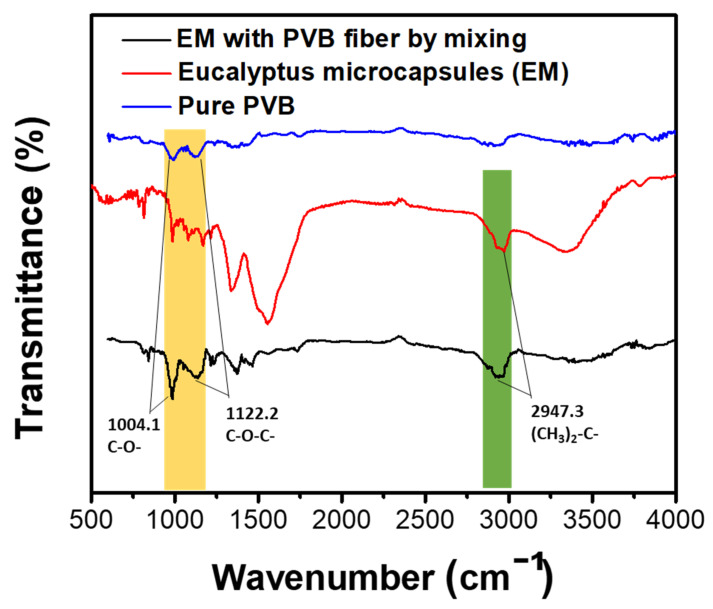
The FTIR picture describes the incorporation of eucalyptus microcapsule within PVB nanofibers fabric. The spectra of pure PVB (blue) and eucalyptus microcapsule (black) confirm the encapsulation of eucalyptus microcapsule into PVB (red) by mixing.

**Figure 4 polymers-14-04447-f004:**
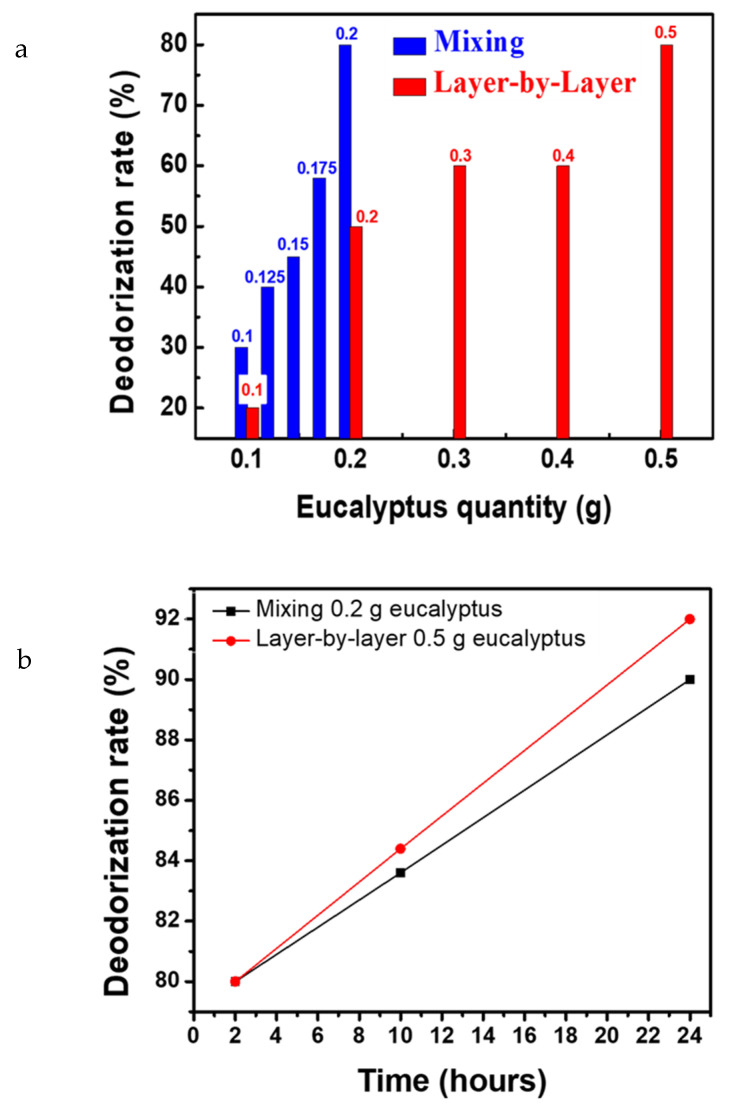
The deodorization rate plot describes the evolution of ammonium content according to the used eucalyptus microcapsule amount. As shown in (**a**), a two-hour deodorization rate test was conducted for layer-by-layer with various eucalyptus microcapsule amount (0.1 to 0.5 g) and mixing with eucalyptus microcapsule amount from 0.1 to 0.2 g. The optimized eucalyptus microcapsule amount was further used to conduct a 24 h deodorization test, as shown in (**b**).

**Figure 5 polymers-14-04447-f005:**
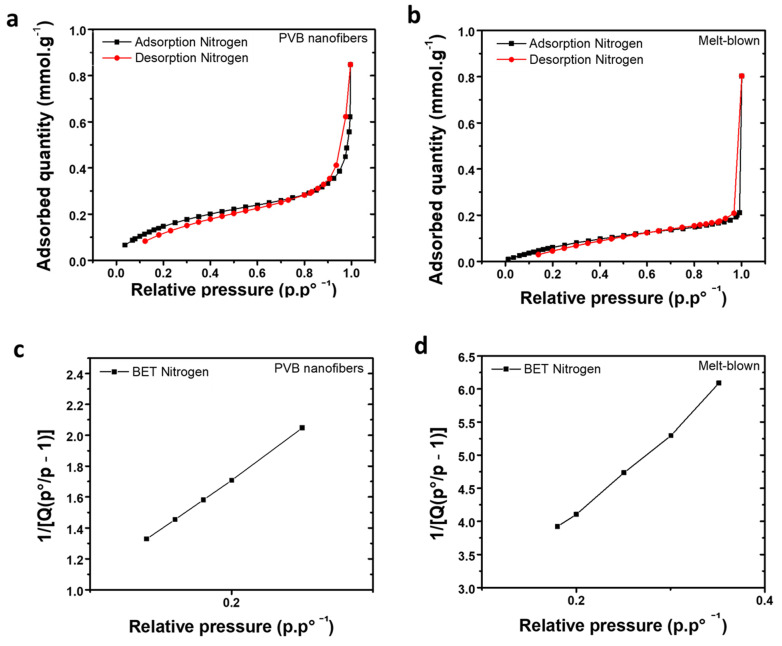
Langmuir adsorption isotherm and BET are plotted for PVB fibers and melt-blown fabric. The (**a**) and (**b**) plot describe PVB and melt-blown fabric Langmuir adsorption isotherm respectively whereas (**c**) and (**d**) describe PVB and melt-blown fabric BET respectively.

**Figure 6 polymers-14-04447-f006:**
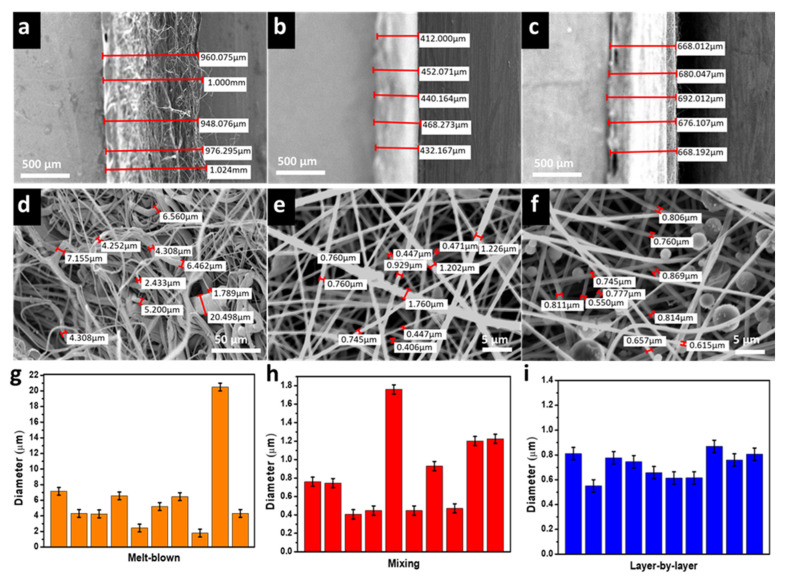
The SEM picture describes the thickness of the melt-blown fabric (**a**), mixing fabric (**b**), and layer-by-layer fabric (**c**). As shown in (**d**,**e**,**f**) each fabric fibers thickness is observed and plotted with their corresponding diameter distribution in (**g**) melt-blown, (**h**) mixing and (**i**) layer-by-layer, respectively.

**Figure 7 polymers-14-04447-f007:**
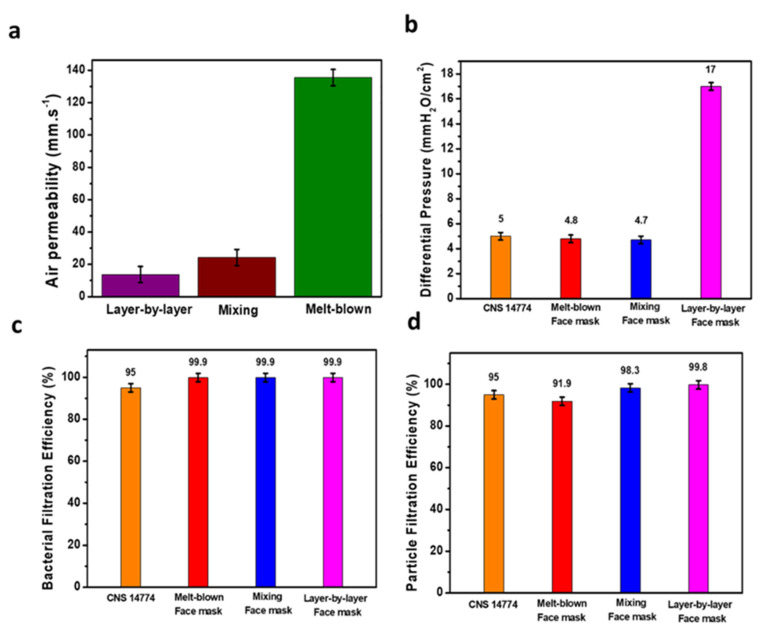
Melt-blown, mixing, and layer-by-layer fabrics are investigated for air permeability (**a**), differential pressure (**b**), Bacterial Filtration Efficiency (**c**), and Particle Filtration Efficiency (**d**) tests. The aim is to put forward the efficiency of PVB–eucalyptus fabric in perspective with the classic melt-blown fabric of the medical mask while assessing their safety and conformability for the CNS 14,774 protocol.

**Table 1 polymers-14-04447-t001:** Evolution of the concentration ammonia in ppm after exposition to eucalyptus-based fabric.

Mixing		Eucalyptus (g)	0	0.1	0.125	0.15	0.175	0.2
	0 h	Initial ammonia concentration (ppm)	100	100	100	100	100	100
	2 h	Final ammonia concentration (ppm)	100	70	60	55	42	20
**Layer-by layer**		**Eucalyptus (g)**	**0**	**0.1**	**0.2**	**0.3**	**0.4**	**0.5**
	0 h	Initial ammonia concentration (ppm)	100	100	100	100	100	100
	2 h	Final ammonia concentration (ppm)	100	80	50	40	40	20

**Table 2 polymers-14-04447-t002:** Comparison of mask fabricated of PVB–eucalyptus microcapsule via mixing and layer-by-layer performances with current medical mask standard.

Mask Type	Air Permeability (mm·s^−1^)	Differential Pressure (mm H_2_O/cm^2^)	Bacterial Filtration Efficiency (%)	Particle Filtration Efficiency (%)
CNS 14,774 General medical face mask	Not required	≦5	≧95	≧80
CNS 14,755 Advanced medical mask	Not required	≦5	≧99.5	≧99.5
Medical face mask (Melt-down	132.3	4.8	99.9	91.9
Our mask (PVB–EM ES mixing)	28.6	4.7	99.9	98.3
Our mask (PVB–EM ES Layer-by-layer)	13.7	17	99.9	99.8

## Data Availability

Not applicable.

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
