# Peer review of "Nanofiber-Based Odor-Free Medical Mask Fabrication Using Polyvinyl Butyral and Eucalyptus Anti Odor Agent"

_polymers, 2022, doi:10.3390/polym14204447_

Round 1

Reviewer 1 Report

This paper describes a nanofiber-based odor-free medical mask fabrication using polyvinyl butyral (PVB) and eucalyptus anti odor agent as the masks’ middle layer. It not only effectively protects against external particle such as coronavirus droplet, but also provides better long-term scents diffusion to increase user’s safety and comfort. I think this paper is suitable for publication in Polymer after addressing the following concerns.

1.      From the SEM images, the PVB/EM nanofiber nonwoven via mixing looks not very uniform, such as the flat areas in the left bottom part of Fig.2d. What is the possible reason? Will the bacterial filtration efficiency be higher if the structure become more uniform?

2.      Error bar is needed for the experimental data in Figure 6 and Figure7.

3.      EM has strong absorption at near 1500 cm-1 in FTIR spectrum, but EM/PVB doesn’t have that peak. What could be the reason?

4.      For the N2 adsorption/desorption isotherm, why the desorption volume is lower than the adsorption volume when the relative pressure is lower than 0.8?

Author Response

Reviewer: 1

Comments to the Author

Reviewer #1: This paper describes a nanofiber-based odor-free medical mask fabrication using polyvinyl butyral (PVB) and eucalyptus anti odor agent as the masks’ middle layer. It not only effectively protects against external particle such as coronavirus droplet, but also provides better long-term scents diffusion to increase user’s safety and comfort. I think this paper is suitable for publication in Polymer after addressing the following concerns.

  1. From the SEM images, the PVB/EM nanofiber nonwoven via mixing looks not very uniform, such as the flat areas in the left bottom part of Fig.2d. What is the possible reason? Will the bacterial filtration efficiency be higher if the structure become more uniform?

Ans: Thank you for your comments. We appreciated that the reviewer provides considerable insight into our manuscript. Usually, the thermodynamic of polymer shows that geometrical confinement of any substance into polymeric chains required to overcome the entropic penalty induced by the presence of the particle. Herein, our EM size is ranging from 3 to 12 µm, therefore the local entropic penalty may result in the displacement of EM at the interface of the PVB or favor the generation of a beads of higher thickness around the EM to accommodate favorably the free energy of the system, hence the reason why some beads shows an elongated or a stacked structure. Therefore, such stacked structure could be attributed to a slightly less uniform solvent evaporation at the EM interfaces, slight deformation of EM capsule as well as the polymeric accommodation around the EM. A more uniform structure would mean that we can perfectly control the size of the EM as well as how the polymer wraps around EM as well as solvent evaporation and PVB stretching. However, the present sample shows highly satisfying result with strong protection and odor suppression. As highlighted in Figure 6, the fiber diameter range at an average of 1 µm, assuming beads would result in thicker fiber, which is much lower than the meltdown fabric, both in raw diameter and shear fiber diameter dispersion. Finally, the bacterial filtration efficiency is already as high as 99.9%, therefore we assume that the presence of larger bead will not have any significant impact.

Herein we add another picture of the sample, we find beads of various size related to the EM polydispersity, however the resulting bacterial filtration efficiency is as high as 99.9% so we believe that a localized modification of active layer structure will not significantly influence the resulting bacterial filtration efficiency. To further shows that our sample is of high quality, the Figure 2d has been replaced as follow, on page 7 of the corrected manuscript:

  1. Error bar is needed for the experimental data in Figure 6 and Figure7.

Ans: Thank you for your comments. We appreciated that the reviewer provides considerable insight into our manuscript. We have added the error bar for Figure 6 and Figure 7. The following Figure 6 and Figure 7 are organized on page 12 and page 13 of the revised manuscript:

  1. EM has strong absorption at near 1500 cm-1 in FTIR spectrum, but EM/PVB doesn’t have that peak. What could be the reason?

Ans: Thank you for your comments. We appreciated that the reviewer provides considerable insight into our manuscript. Usually, EM is made of melamine and the corresponding signal at 1500 cm-1 in FTIR belongs to the triazine rings whereas the signal at 1335.4 cm-1 belongs to triazine amine groups. We find that upon encapsulation, all peaks associated with EM are still present but highly weakened, indicating a strong encapsulation of the EM by PVB. Considering that the triazine peaks at 1500 cm-1 in the FTIR plots has almost disappeared, it may be attributed to encapsulation or interaction with PVB through triazine. However, the presence of the 2947.3 cm-1 signal peak associated to the 1,8-cineole is still present, indicating that the oil release will not be affected and the agent primary role will be sustained. To further strengthen our argumentation, the following has been added, on page 7 of the corrected manuscript from line 269 to line 278:

“The eucalyptus microcapsule, in which the shell is composed of melamine, FTIR analysis is given by 1335.4 cm-1 and 3340.4 cm-1 peaks ascribed to melamine amine group whereas 814.0 and 1559.2 cm-1 signal peaks are associated with the 1, 3, 5-triazine ring. Finally, the 2947.3 cm-1 signal peak is associated to the 1,8-cineole, the main component of the eucalyptus essential oil [49–51]. Further, mixing approach PVB-Eucalyptus materials were analyzed by FTIR, the 1004.1 cm-1 C-O and 1139.1 cm-1 C-O-C stretching characteristic of PVB are visible. Upon encapsulation, all peaks associated with the melamine are still present but weakened, indicating a strong encapsulation of the eucalyptus microcapsule by PVB. The triazine peaks at 1500    cm-1 in the FTIR plots has almost disappeared and may be attributed to interaction with PVB.”

  1. For the N2 adsorption/desorption isotherm, why the desorption volume is lower than the adsorption volume when the relative pressure is lower than 0.8?

Ans: Thank you for your comments. We appreciated that the reviewer provides considerable insight into our manuscript. In general, the adsorption process is pressure dependent, meaning that as the relative pressure increase, the adsorbed volume increases. Furthermore, physisorption is a physical process and the isotherm behavior is intrinsically linked with the material surface nature. For instance, a type I isotherm is related to microporous material with adsorption occurring through activated carbon while a type V isotherm fit mesoporous materials such as PVB. Hysteresis have been observed at higher pressure, but at lower pressure, it may be attributed various factor such as acceleration of desorption of gas, to a discrepancy in the product concentration and experimental parameters. However, we believe it does not impact significantly the role of PVB and that the reflected behavior still indicates a proper desorption of N2 alongside a greater surface area for PVB than the melt-down fabric.

Reviewer 2 Report

The article studied the production of medical masks from polyvinyl butyral nanofibers with the addition of eucalyptus anti odor agent by electrospinning. The article is interesting and useful. There are some comments:

- Specify the characteristics of PVB.

- Write Name and company/country of spinning equipment.

- There are not enough spaces in the text before the links.

- In Figure 6, the red font is hard to see.

- Do not need a dot at the end of the title.

- Give a table comparing the performance of your mask and an existing standard.

Author Response

Reviewer: 2

Comments to the Author

Reviewer #2: The article studied the production of medical masks from polyvinyl butyral nanofibers with the addition of eucalyptus anti odor agent by electrospinning. The article is interesting and useful. There are some comments:

- Specify the characteristics of PVB.         

Ans: Thank you for your comments. We appreciated that the reviewer provides considerable insight into our manuscript. Polyvinyl butyral is a commonly used polymer for electrospinning based on its friendly properties such as antibacterial capabilities, non-toxicity porosity, hydrophobicity and environmentally friendly properties. Furthermore, PVB is a highly porous materials, enabling to host numerous pore that will be suitable to undergo strong and repetitive adsorption-desorption process. Finally, PVB exhibit favorable surface tension as well as an acceptable conductivity and can be dissolved in polar solvent without the use of conductive additive. Therefore, PVB characteristics have been added in the introduction and the justification of PVB choice was developed in the section 3.1. To further strengthen our argumentation, the following has been added, we refer to the following changes on page 3 (Introduction) from line 104 to line 107 and page 5 from line 223 to line 228 (Section 3.1):                                                             

Herein, we use PVB due to its non-toxicity, porosity, hydrophobicity and environmentally friendly properties, combined with eucalyptus to develop two encapsulations approach namely mixing and layer-by-layer. PVB high porosity enable highly efficient molecule absorption and desorption as well as antibacterial performances. PVB is widely used in ink printing adhesives on food packaging in Europe and the United States as well as mask fabrication, ascribed to its remarkable flexibility and adhesion properties. Moreover, PVB has non-toxic and hydrophobic properties preventing the water vapor and the generated saliva from triggering the membrane mask degradation. The flexibility of PVB makes it a suitable candidate for electrospinning due to its high conductivity and surface tension.

- Write Name and company/country of spinning equipment.

Ans: Thank you for your comments. We appreciated that the reviewer provides considerable insight into our manuscript. The following has been added on page 3 from line 129 to line 135.

“Stainless steel needle electrospinning equipment specifications are as follow: No.18: inner diameter: 0.96 mm, outer diameter: 1.26 mm; length: 25.4 mm. No.22: inner diameter: 0.42 mm, outer diameter: 0.72 mm; length: 2.54 mm. All needles have been supplied by Hong Zhun Enterprise Co., Ltd (Taiwan). The needle is placed at a vertical position on the electrospinning set-up with the following: high voltage power supply (manufacturer SIMCO Industrial (Taiwan)), Static Control Regulator (Manufacturer: LOKO power (Taiwan)) and the injection pump (Manufacturer: kd Scientific (United States))”

- There are not enough spaces in the text before the links.

Ans: Thank you for your comments. We appreciated that the reviewer provides considerable insight into our manuscript. We have modified the spacing in the text right before the DOI in the reference as well as in the main manuscript before the referencing.

- In Figure 6, the red font is hard to see.

Ans: Thank you for your comments. We appreciated that the reviewer provides considerable insight into our manuscript. We have modified the font as below:    The following Figure 6 is organized on page 12 of the revised manuscript:

- Do not need a dot at the end of the title.

Ans: Thank you for your comments. We appreciated that the reviewer provides considerable insight into our manuscript. Titles of section and subsection dots have consequently been removed.

- Give a table comparing the performance of your mask and an existing standard.

Ans: Thank you for your comments. We appreciated that the reviewer provides considerable insight into our manuscript. We have added the following table as a table comparing our mask performances with the CNS 14774 (General medical face mask) and CNS 14755 (Advanced medical face mask) standards (Taiwanese specifications). The following description and Table 2 are organized on page 14 and page 15 of the revised manuscript:

 “Our mask performances have been summarized in the Table 2 to emphasize their promising capabilities in comparison with current medical mask standard.”

Table 2. Comparison of mask made of PVB – eucalyptus microcapsule via mixing and layer-by-layer performances with current medical mask standard.

Mask type

Air permeability (mm.s-1)

Differential pressure (mmH2O/cm2)

Bacterial filtration efficiency (%)

Particle filtration efficiency (%)

CNS 14774 -General medical face mask

Not required

≦5

≧95

≧80

CNS 14755 – Advanced medical mask

Not required

≦5

≧99.5

≧99.5

Medical face mask (Melt-down

132.3

4.8

99.9

91.9

Our mask (PVB – EM ES mixing)

28.6

4.7

99.9

98.3

Our mask (PVB – EM ES Layer-by-layer)

13.7

17

99.9

99.8
